# RNA Recognition and Immunity—Innate Immune Sensing and Its Posttranscriptional Regulation Mechanisms

**DOI:** 10.3390/cells9071701

**Published:** 2020-07-16

**Authors:** Takuya Uehata, Osamu Takeuchi

**Affiliations:** Department of Medical Chemistry, Graduate School of Medicine, Kyoto University, Yoshida-Konoe-cho, Sakyo-ku, Kyoto 606-8501, Japan; tuehata@mfour.med.kyoto-u.ac.jp

**Keywords:** RNA virus, Toll-like receptors, RIG-I-like receptors, type I interferons, cytokines, RNA binding proteins, Regnase-1, mRNA decay

## Abstract

RNA acts as an immunostimulatory molecule in the innate immune system to activate nucleic acid sensors. It functions as an intermediate, conveying genetic information to control inflammatory responses. A key mechanism for RNA sensing is discriminating self from non-self nucleic acids to initiate antiviral responses reliably, including the expression of type I interferon (IFN) and IFN-stimulated genes. Another important aspect of the RNA-mediated inflammatory response is posttranscriptional regulation of gene expression, where RNA-binding proteins (RBPs) have essential roles in various RNA metabolisms, including splicing, nuclear export, modification, and translation and mRNA degradation. Recent evidence suggests that the control of mRNA stability is closely involved in signal transduction and orchestrates immune responses. In this study, we review the current understanding of how RNA is sensed by host RNA sensing machinery and discuss self/non-self-discrimination in innate immunity focusing on mammalian species. Finally, we discuss how posttranscriptional regulation by RBPs shape immune reactions.

## 1. Introduction

Innate immunity is a first-line host defense mechanism that initiates inflammatory responses against pathogen invasion. The main characteristic of initial innate immune responses is the production of inflammatory cytokines and chemokines, which leads to inflammatory cell activation and recruitment. Innate immune cells are genetically equipped with pathogen sensors, which are called pattern recognition receptors (PRRs). These PRRs recognize not only pathogen-associated molecular patterns (PAMPs) but also endogenous molecules, which are called damage-associated molecular patterns (DAMPs) [1,2,3,4]. Toll-like receptors (TLRs) are one of the best-characterized PRRs on cellular membranes, with 11 transcripts/10 proteins in human TLRs. TLRs are evolutionally conserved among species [5,6]. They are involved in recognizing a broad range of pathogens, including bacteria, viruses, and fungi [7,8]. RIG-I-like receptors (RLRs) and NOD-like receptors are cytoplasmic receptors that contribute to inflammatory responses via the activation of downstream signaling pathways [9,10,11]. Although various myeloid lineages participate in the detection of pathogens through these PRRs, dendritic cells (DCs) play the vital role of bridging innate and adaptive immune responses. Furthermore, the activation of PRR signaling pathways primes DCs that undergo maturation and results in the expression of a set of costimulatory molecules. These DCs efficiently present exogenous antigens to antigen-specific CD4 T cells on MHC-II, which leads to the specification of helper T cell differentiation depending on microenvironmental cues [2]. PRRs also enhance MHC-I-mediated presentation of exogenous antigens to cytotoxic T lymphocytes, a process called cross-presentation [12,13]. PRRs’ roles have been described not only in infection but also in many other pathological conditions, including cancer, metabolic disorders, and aging [14,15,16].

RNA and its recognition are critical for regulating inflammatory processes at multiple steps. Host RNA sensors recognize foreign RNAs as well as endogenous RNAs from stressed cells. Furthermore, there are numerous RNA-binding proteins (RBPs) that modulate immune reactions by controlling gene expression [17]. Posttranscriptional regulation refers to the mechanisms governing the messenger RNA (mRNA) life cycle that includes splicing, nuclear export, mRNA modification, translation, and mRNA decay. Throughout the life cycle, mRNA and RBPs assemble to develop ribonucleoproteins (RNPs) as a functional unit and form a complex network of RBP–RNA interaction that depends on the cellular context [18]. RBPs play a central role in determining mRNA fate and contributing to inflammatory responses through viral RNA recognition and various RNA metabolisms. In this study, we review nucleic acid-sensing mechanisms in innate immunity, with a particular focus on viral and other endogenous RNAs. We also discuss the mechanisms that regulate these responses at posttranscriptional levels by RBPs (for extensive review of non-coding RNA-mediated posttranscriptional regulation, see [19,20,21]).

## 2. RNA Sensing Mechanisms

The innate immune system has nucleic acid sensors that keep non-self nucleic acids under constant surveillance. Among human TLRs, TLR3, TLR7, and TLR8 are endosomal RNA sensors, whereas TLR3 recognizes double-stranded (ds) RNA. In mice, TLR8 is dispensable for RNA recognition, which completely depends on TLR7. TLR7 and TLR8 sense single-stranded (ss) RNA, whereas TLR13 expressed in mice also recognizes ssRNA derived from bacterial 23S ribosomal RNA [22,23]. RLRs consist of RIG-I, MDA5, and LGP2, which are characterized by a DEAD-box helicase domain and the C-terminal dsRNA-binding domain. RIG-I and MDA5 also harbor caspase-recruitment domains (CARDs) that are responsible for the activation of signaling pathways following the recognition of distinct dsRNA features [24]. TLR9 and cyclic guanosine monophosphate (GMP)–adenosine monophosphate (AMP) synthase (cGAS) act as endosomal and cytosolic DNA sensors that recognize unmethylated CpG containing ssDNA and dsDNA, respectively [25,26]. After ligand engagement, these sensors activate signaling pathways that induce type I IFNs to initiate antiviral responses. TLRs and RLRs also trigger signal pathways and activate shared downstream signaling modules. For example, TLRs (except TLR3) activate IκB kinases (IKKs) and MAPKs downstream of adaptor protein MyD88. These signaling modules then activate NF-κB and AP-1, which induce inflammatory cytokines. Upon ligand stimulation, TLR3 and TLR4 recruit Toll/IL-1R domain-containing adaptor-inducing IFN-β (TRIF), which is another adaptor protein. Subsequently, TRIF activates TANK-binding kinase 1 (TBK1) and IKK-i/ε, which phosphorylate IRF3 and IRF7. This phosphorylation induces their dimerization and nuclear translocation to transactivate the expression of type I IFN genes. MyD88 signaling downstream of TLR 7-9 also induces type I IFN genes via IRF5 and IRF7 [27,28]. On the other hand, RLRs associate with MAVS/IPS-1, which is a key adaptor protein localized on the mitochondrial membrane, and this activates IRF3/7 via the TBK1-IKK-i/ε and induces type I IFN responses.

Type I IFNs have multiple functions in the fight against viral infection (Figure 1). Type I IFNs, along with IFN-stimulated genes (ISGs), promote apoptosis of infected cells and confer resistance to viral infection in surrounding uninfected cells [29]. Simultaneously, Type I IFNs induce DC maturation, activate virus-specific cytotoxic CD8 T cells, and enhance the NK cell cytotoxicity. Secreted type I IFNs act on the interferon-α/β receptor (IFNAR), which activates the Janus kinase–signal transducer and the transcription (JAK–STAT) pathway, leading to ISG expression [30]. These sequential events lead to the subsequent activation of adaptive immune cells, characterized by the differentiation of CD4 T cells into a particular subset of helper T cells and antigen-specific antibody production by B cells.

According to extensive studies, these sensors not only survey pathogen invasion but also recognize host nucleic acids. Immune responses elicited by endogenous nucleic acids are independent of pathogen infection, a process known as sterile inflammation [31]. Several studies have implicated that the spontaneous activation of host nucleic acid sensors in autoimmune diseases, cancer, and other chronic inflammatory pathologies [16,32,33]. 

## 3. Elements for Discrimination between Self and Non-Self RNA

Because self RNAs are distinguished from foreign RNAs in nature, host cells have evolved multiple molecular mechanisms used in the discrimination of self and non-self RNAs. Long dsRNAs in the cytosol are derived from the genome of RNA viruses or intermediate products that are generated during viral replication; they are undetected in healthy cells [34]. Therefore, dsRNAs are regarded as non-self and act as ligands for RNA sensors. Polyinosinic–polycytidylic acid (Poly(I:C)), which is a synthetic dsRNA analog, is a typical ligand for TLR3 and RLRs and can highly induce type I IFNs (Figure 2a). In addition to long dsRNAs, TLR3 can also recognize short dsRNAs with minimum lengths >35 bp [35].

Among RLR family members, RIG-I recognizes short dsRNAs. For the optimal activation of RIG-I, dsRNAs require: (1) A triphosphate or diphosphate group on the 5’ end of RNA molecules [36,37], (2) the absence of 2′-O-methylation of the 5′ end of nucleotides [38], and (3) blunt-ended base paring (Figure 2b). The dsRNA length must be 19 bp or longer to induce RIG-I-mediated type I IFNs [36,39]. RIG-I can also recognize diphosphate structures at 5’ end that are produced by some viruses (e.g., reovirus) through its triphosphatase [40]. Since poly (I:C) also contains 5’diphosphate moieties, the short fragment of poly (I:C) can activate RIG-I [40]. However, the host mRNAs are single-stranded, 5′-capped, and 2′-O-methylated. This characteristic enables host mRNAs to escape from self-recognition, even though they bear triphosphate at the 5’ end. Other RNA species, such as ribosomal RNA and transfer RNA, do not bear the 5′-capping structure but are instead processed into the 5’-monophosphate form [41]. These characteristics contribute to the discrimination of self/non-self. In contrast, MDA5 recognizes long dsRNA >300 bp [42,43,44], although the minimum requirement for length is still unclear. After dsRNA recognition, MDA5 assembles into filaments along with RNA duplexes, which leads to CARD oligomerization [45,46,47]. This oligomerization promotes the microfibril formation by CARD of MAVS. Cryoelectron microscopy (Cryo-EM) showed that MDA5 structures are different twist forms of MDA5-dsRNA, and each MDA5 molecule binds to 14 or 15 bp depending on these twist forms [48]. MDA5 typically recognizes dsRNA generated from (+)ssRNA viruses (e.g., picornavirus) during viral replication in the cytosol [43]. In addition, (–)ssRNA viruses, which do not produce dsRNA, can induce type IFN via MDA5 [34]. This characteristic suggests that MDA5 recognizes secondary structures as RNA duplexes.

TLR7 and TLR8 recognize ssRNA [49,50] (Figure 2a). These TLRs preferentially recognize polyuridine (polyU) and guanosine/uridine-rich (GU-rich) sequences via leucine-rich repeats (LRRs) [51]. TLR7 and TLR8 also recognize RNA degradation products and require free guanosines and uridines, respectively, for maximum activation [52,53,54]. TLR7 and TLR8 share ligand-binding mechanisms where there are two distinct RNA-binding sites in the M-shaped dimer that form symmetrically side-by-side. The first binding site is at the dimerization interface, conserved between TLR7 and TLR8, and binds free nucleoside, guanosine, or uridine. In contrast, the second binding pocket, which is on the concave surface of LRRs, binds short oligonucleotides. Recent work showed that TLR8 signaling requires a lysosomal endonuclease RNaseT2 and RNase2, which generates uridines by cleaving exogenous ssRNAs before and after the uridine residue [55,56]. This mechanism partially explains the reason why GU-rich sequences are important for the recognition of TLR8, although the sequences are also present in self RNAs. Besides, the compartmentalized action of RNaseT2 in the lysosomal acidic condition makes TLR8 specific for recognizing foreign RNAs. It remains to be elucidated how guanosine molecules are generated for triggering TLR7 signaling.

## 4. Recognition of Endogenous RNAs in Diseases

Increasing evidence demonstrates that host RNA-sensing mechanisms are also potent in recognizing endogenous RNAs under pathological conditions. Genetic mutations of RNA sensors and abnormal RNA metabolism cause pathologies, which are induced by aberrant activation of type I IFNs called type I interferonopathies. Examples of these are Singleton–Merten syndrome (SMS) and Aicardi–Goutieres syndrome (AGS). The gain of function mutations of DDX58 and IFIH1, which encode RIG-I and MDA5, respectively, are reported in studies on whole-exome sequencing in patients with Singleton–Merten syndrome (SMS) and Aicardi–Goutieres syndrome (AGS) [57,58,59,60]. These mutations may lead to higher production of type I IFNs through increased RNA avidity. N-ethyl-N-nitrosourea (ENU) mutagenesis identified an MDA5 mutation in mice and showed spontaneous autoimmunity development without RNA recognition [61]. This G821S mutant highly expressed IFNs without any stimuli, which suggests RNA-independent auto-activation of MDA5. In addition to the mutation of RLRs, type I interferonopathies are also caused by the accumulation of aberrant nucleic acids through the failure of RNA metabolism. Mutations in an A-to-I editing enzyme, dsRNA-specific adenosine deaminase (ADAR) 1, are predisposed to AGS development [62]. Where an ADAR1 deficiency in mice resulted in embryonic lethality due to an aberrant type I IFN production and failed erythropoiesis [63], this phenotype partly depended on the MDA5–MAVS pathway activation [64,65]. Therefore, ADAR1 edits endogenous dsRNAs to prevent recognition by MDA5. Alu retroelements form ~300 bp Alu:Alu duplex RNAs when they are in an inverted repeat configuration and nearby [66,67]. Alu is a major target of ADAR1-mediated A-to-I editing. Therefore, compromised ADAR1 function or constitutive activation of MDA5 in AGS patients results in the triggering of type I IFN responses by aberrant recognition of Alu elements [67]. Immune stimulatory endogenous dsRNA is also potently generated by bilateral transcription of mitochondrial DNA [68]. Mitochondrial degradosome components, including RNA helicase SUV3 and polynucleotide phosphorylase (PNPase), suppress MDA5-mediated recognition of mitochondrial dsRNAs via the degradation of L-strand RNA. Consistently, patients harboring a hypomorphic mutation in PNPT1 (which encodes PNPase) showed ISG upregulation. Thus, host RNAs are metabolized elaborately to prevent aberrant recognition by innate immune sensors culminating type I IFN responses (Figure 3).

On the other hand, aberrant activations of TLR7 and TLR9 have been implicated in type I IFN-related autoimmune diseases, such as systemic lupus erythematosus (SLE), which is characterized by the presence of autoantibodies against nucleic acids. These two TLRs are expressed particularly in plasmacytoid DCs (pDCs) and B cells. The roles of TLR7 and TLR9 in the pathogenesis of SLE have been shown to be quite different. The severity of the disease depends on the strength of TLR7 signaling. Overexpression of TLR7 in mice accelerates the development of SLE, whereas deficiency of TLR7 in lupus-prone mice ameliorates the phenotype [69]. Mice harboring the *Yaa* chromosome are used as a male-specific lupus model in which IFN responses are enhanced due to TLR7 duplication [70,71]. However, TLR9 signaling is protective against the development of SLE. TLR9 deficiency exacerbates the disease. Nevertheless, TLR9 contributes to the production of an anti-dsDNA antibody [72]. In addition, overexpression of TLR9 in mice is not sufficient for autoimmunity [73]. The differential roles of these two TLRs are still being investigated. A possible explanation may be their differential mechanisms through the trafficking chaperone Unc93B1. Unc93B1 associates with TLR9 more preferentially than TLR7, which is essential for preventing aberrant TLR7 activation. D34A mutation of Unc93B1 causes autoimmunity due to the imbalance toward TLR7 [74]. Once transported into endosomes, TLR9 dissociates with Unc93B1, while TLR7 still interacts with Unc93B1 [75,76]. This sustained interaction with Unc93B1 is important for subsequent recruitment of syntenin-1, which is involved in exosome biogenesis [75]. This, in turn, leads to termination of TLR7 signaling through further trafficking to intraluminal vesicles of multivesicular bodies. A mutation of Unc93B1 that fails to recruit syntenin-1 causes lupus-like phenotype [75]. Therefore, multiple mechanisms that restrict TLR7 activation are required for immune homeostasis.

## 5. RBPs Directly Suppress Viral RNAs

Type I IFN signaling inhibits viral replication by inducing a large array of genes. For instance, IFN-inducible RBPs play critical roles in antiviral defense by binding viral RNAs (Figure 2c). Protein kinase R (PKR), which is an IFN-inducible dsRNA-activated protein kinase, undergoes dimerization and auto-phosphorylation upon ligation of viral dsRNA. PKR activation inhibits cap-dependent translation via phosphorylation of eIF2α [77,78]. PKR is also critical for the formation of stress granules elicited by viral infection where RIG-I is recruited and initiates signaling pathways to induce type I IFNs [79]. Recent work shows that endogenous circular RNA, which harbors short dsRNAs, binds PKR and controls its activation. However, in SLE patients, the reduction of circular RNAs leads to aberrant PKR activation [80]. IFN-induced proteins with tetratricopeptide repeats are another example of ISGs harboring antiviral activity by binding with viral RNAs. IFIT1 recognizes viral RNAs, such as those lacking 2′-O-methylation on their 5′ 7-methylguanosine cap (m7G cap) structure or RNAs with a 5′-triphosphate end [81,82], and inhibits translation and sequesters viral RNA from replication [83]. ADAR1 is also induced upon viral invasion and binds to viral dsRNA. However, the role of ADAR1 against viral replication is still controversial. Previous studies show that A-to-G mutation in viral genomes is observed [84,85], which suggests the involvement of ADAR1 in viral replication. Yet, evidence that the editing itself restricts viral replication is lacking. Additionally, ADAR1 suppresses antiviral responses by blocking cytoplasmic RNA sensing pathways [65,86,87]. Further studies will elucidate the precise mechanisms via RNA editing.

Host RNases also restrict viral infection by directly degrading viral RNAs. RNase L, which is a cytoplasmic endoribonuclease, degrades viral RNA molecules. The activation of RNase L requires dimerization via 2′-5′-oligoadenylate that is synthesized by 2′-5′-oligoadenylate synthetases (OASs) upon recognition of dsRNAs [88]. RNase L also degrades endogenous circular RNAs in the context of viral infection, which results in PKR activation through the mechanism mentioned above [80]. In the nucleus, another IFN-inducible endoribonuclease, NEDD4-binding protein 1 (N4BP1), restricts human immunodeficiency virus-1 (HIV-1) replication by degrading HIV-1-derived transcripts in T cells and macrophages [89]. The sequence specificity against HIV RNA requires clarification.

## 6. Posttranscriptional Mechanisms Regulating Host Inflammatory Gene Expression

As previously discussed, nucleic acid recognition by PRRs induces type I IFNs and inflammatory cytokines to combat viral infection. Such inflammatory genes are post-transcriptionally regulated by RBPs to control the magnitude and duration of immune responses. In addition, increasing evidence shows that posttranslational modifications, in response to PRR stimulation, tightly regulate RBP function [90,91]. The half-lives of mRNAs encoding inflammatory cytokines are much shorter than house-keeping genes, which indicates that mRNA decay machinery is critically involved in regulating inflammatory responses [92]. In general, mRNA decay mechanisms are divided into two distinct pathways: Exo- and endo-nucleolytic [93]. The 5′ cap structure and poly(A) tail protect mRNAs from degradation, and deadenylation in the exonucleolytic pathway initiates mRNA decay. In this step, deadenylases, including PARN and PAN2-PAN3 and the CCR4-NOT complex, contribute to the sequential shortening of the poly A tail. Then, deadenylation promotes the removal of the 5′ cap by decapping enzymes, DCP1 and DCP2, followed by 5′-to-3′ and 3′-to-5′ mRNA degradation by XRN1 and exosome complexes, respectively. Alternatively, mRNAs undergo endonucleolytic cleavage by endoribonucleases, including SMG6 or Regnase-1, which is followed by exonucleolytic degradation. The selection of the decay mechanism depends on RNP compositions and their interactions with cis-regulatory elements of mRNA.

An AU-rich element (ARE) in mRNA 3’ UTR is one of the best-studied cis-elements and acts as a code for mRNA decay, translation control, and nuclear export (Figure 4a). ARE is typically represented as the core sequence of the AUUUA pentamer and forms ARE clusters in an AU-rich context [94]. In the genome, ARE-containing transcripts are commonly observed and comprise 5–22% of human mRNAs [95,96]. AREs are recognized by a set of ARE-binding proteins such as tristetraprolin (TTP, also known as ZFP36). TTP harbors tandem CCCH-type zinc finger domains essential for the binding of ARE, and recruits the CCR4-NOT complex and the decapping complex Dcp1 and Dcp2 to trigger exonucleolytic mRNA decay [97,98,99,100] (Figure 4b). The TTP family proteins, ZFP36L1 and ZFP36L2, similarly bind to mRNAs and exert their functions (although their biological roles are different) [101]. Other ARE-binding proteins, such as AUF1 (also known as hnRNPD), KH-type splicing regulatory protein (KSRP), and ELAVL1 (also known as HuR), also exist. In contrast to TTP and other ARE-biding proteins, ELAVL1 is an ARE-binding protein that stabilizes mRNAs. This protein predominantly localizes in the nucleus but translocates into the cytoplasm in response to viral infection, hypoxic stress, and DNA damage [102,103,104]. Subsequently, ELAVL1 competes against other ARE-binding proteins for ARE sites and therefore enhances the stability and translation of target mRNA.

Stem-loop structures in inflammatory mRNAs also act as a cis-element recognized by RBPs such as Regngase-1 and Roquin (Figure 4c). These RBPs cause the interaction of stem-loop structures with a pyrimidine-purine-pyrimidine loop sequence [105,106]. Regnase-1 localizes with the endoplasmic reticulum (ER) or cytoplasm and associates with ribosomes. Regnase-1 degrades target mRNAs following the pioneer rounds of translation [107]. Regnase-1 is associated with UPF1, which is a helicase that is essential for non-sense-mediated mRNA decay. After translation terminates, Ser/Thr protein kinase SMG1 phosphorylates UPF1, which promotes unwinding of stem-loop structures through its helicase activity and enables Regnase-1 to degrade target mRNA. Roquin is a distinct RBP that recognizes the stem-loop structure through its ROQ domain [108,109]. Where the binding motives of Roquin and Regnase-1 overlap, Roquin localizes to stress granules and P bodies, and induces mRNA degradation through the recruitment of deadenylation and decapping complexes [105,110]. Regnase-1 and Roquin control inflammatory gene expression in the acute and more chronic phases of inflammation, respectively [106]. Thus, RBPs that share binding motives function spatiotemporally and fine-tune inflammatory responses. mRNAs that undergo tight regulation tend to harbor more than one cis-element where a complex interaction with multiple RBPs forms to regulate them. For example, the stability of Tnf mRNA is regulated via two distinct elements, which include both ARE and the stem-loop structure, by TTP and Roquin.

## 7. The Role of Posttranscriptional Regulation in Immune Homeostasis

Posttranscriptional control in the innate immune system promptly induces inflammatory responses and simultaneously produces a timely resolution of inflammation. Dysregulation of posttranscriptional mechanisms, especially those involved in mRNA stability and decay, causes sustained inflammation, and leads to autoimmunity. In particular, the biological roles of ARE-binding proteins have been studied extensively. For example, TTP is a well-characterized RBP that is essential for the maintenance of immune homeostasis. Furthermore, a TTP deficiency in mice causes spontaneous development of arthritis and other inflammatory syndromes, such as dermatitis and cachexia, which are mainly due to the overproduction of TNF [111,112]. Consistently, ARE deletion in Tnf mRNA also shows a similar phenotype in mice [113], which highlights the importance of TTP-mediated posttranscriptional regulation of the Tnf transcript for maintenance of immune homeostasis. TTP regulates various transcripts that encode proinflammatory cytokines including Il6, Csf2, and Ifng [114,115,116,117,118]. Anti-inflammatory genes, such as Il10, are also targeted by TTP [119]. In macrophages, TTP is expressed constitutively before activation. Once TLR signaling is triggered, TTP is phosphorylated by MAPK-activated protein kinase 2 (MK2) downstream of the p38 mitogen-activated protein kinase (MAPK) [120,121]. The phosphorylated form of TTP is bound to 14-3-3 protein, which inhibits the recruitment of the deadenylation complex CCR4-NOT1 [121,122] (Figure 4b). The significance of this mechanism was previously verified in mice harboring mutations at phosphorylation sites and showed that the non-phosphorylatable form of TTP potently suppresses inflammatory responses [123]. This phosphorylation can be reversed by two phosphatases, protein phosphatase 2A (PP2A) and MAPK phosphatase 1 (MKP1), which suggests that the balance between phosphorylated and unphosphorylated forms of TTP is essential for the control of inflammation [124,125].

ZFP36L1 and ZFP36L2 play redundant roles in lymphocyte development. During early lymphopoiesis, ZFP36L1 and ZFP36L2 regulate mRNA-encoding cell cycle regulators [126,127]. The conditional deletion of both genes in mice causes severely impaired lymphocyte development, which eventually leads to T lymphoblastic leukemia [126]. It is still unclear what roles ZFP36L1 and ZFP36L2 play in myeloid cells. Mice specifically lacking ZFP36L1 in myeloid cells do not show differential responses to acute bacterial pneumonia, although ZFP family genes might function redundantly [128]. Nevertheless, ZFP36L1 regulates senescence-associated secretory phenotype (SASP) by destabilizing Il1b, Il8, and Cdkn1a [129]. In this context, MK2 also phosphorylates ZFP36L1, which leads to proteasomal degradation. This control of ZFP36L1 expression is involved in the mTOR pathway, where the translation of MK2 is enhanced through eukaryotic translation initiation factor 4E (EIF4E)-binding proteins (4EBPs). The non-phosphorylatable mutant of this protein can reduce the expression of SASP transcripts. Notably, similar mutants of TTP and ZFP36L2 show a partial reduction in senescence, which suggests that there are redundant mechanisms for SASP progression.

AUF1 is another ARE-binding protein involved in mRNA destabilization. Among four isoforms, consisting of p37, p40, p42, and p45, p37 profoundly influences mRNA stability [130]. AUF-deficient mice show chronic inflammation and a profound endotoxin shock in response to LPS [131,132], and systemic inflammation in these mice is attributed to the overproduction of TNF and IL-1β. KSRP is also required for ARE-mediated mRNA decay. This protein destabilizes mRNA encoding type I IFN, such as Ifna and Ifnb [133]. Additionally, Ksrp-deficient mice are resistant to HSV-1 infection. The role of ELAVL1 in the immune system is much more complicated. In macrophages, HuR binds to Tnf mRNA to promote stabilization [134]. HuR is also involved in viral responses through the activation of RLR. HuR promotes IRF3 nuclear translocation via the increased stability of Plk2 mRNA [104]. In contrast, mice specifically lacking ELAVL1 in macrophages show exacerbated inflammatory responses in chemically induced colitis [135]. Thus, these contradictory findings indicate the diverse functions of ELAVL1 [136,137,138,139].

Stem-loop-binding proteins are also involved in the maintenance of immune homeostasis. Regnase-1 is expressed ubiquitously, and this protein plays an essential role in the immune system. This protein’s importance has been demonstrated in mice lacking Regnase-1, which shows diverse inflammatory phenotypes, including lymphadenopathy, massive infiltration of lymphocytes in the lungs and liver, and the production of autoantibodies [140]. In macrophages, Regnase-1 destabilizes a set of genes encoding inflammatory cytokines, including Il6, Il12p40, and Il1b. Thus, Regnase-1-deficient macrophages show marked production of these inflammatory cytokines [140]. Regnase-1-mediated mRNA destabilization is antagonized by Arid5a, which is another RBP [141,142]. Upon stimulation with LPS, Arid5a translocates from the nucleus to the cytosol and binds to 3’UTR of Il6 mRNA; this process protects target mRNAs from degradation [143]. After LPS stimulation, Regnase-1 mRNA is induced rapidly. In sharp contrast, the Regnase-1 protein undergoes IKK-mediated phosphorylation and subsequent proteasomal degradation, which is reminiscent of IκBα [144] (Figure 4c). Indeed, Regnase-1 is phosphorylated at the DSGXXS motif, which is present in its C-terminal domain. This degradation depends on MyD88 but not TRIF. Despite the engagement of the IKK complex, the TNF pathway is not involved in the change of Regnase-1 protein expression. The IL-17-mediated Regnase-1 phosphorylation regulatory mechanism is also important in controlling its function. The TBK1/IKK-i/ε complex induces phosphorylation and leads to the dissociation of Regnase-1 oligomerization and translocation from the ER/polysome to the cytosol [145]. In accordance with this idea, Regnase-1 is involved in the remodeling of the epithelial tissue of patients who have ulcerative colitis (UC). An exome sequencing study showed that ZC3H12A mutation in Ser438, which is present within the DSGXXS motif, is highly enriched in the inflamed tissue of UC samples [146,147]. This mutant form of Regnase-1 is resistant to degradation induced by IL-17-mediated IKK activation and indicates that Regnase-1 suppresses chronic inflammation in the UC epithelium. This is consistent with another study showing knock-in mice harboring mutations at Ser435 and Ser439 (Regnase-1S435A/S439A), and EAE development is ameliorated by the failure of non-hematopoietic cells to induce inflammatory responses [145]. Regnase-1 is also essential in controlling the activation of adaptive immune cells. For example, mice lacking Regnase-1, specifically in T cells, recapitulate the phenotype of Regnase-1 null mice [148]. Regnase-1 targets various genes, including Il2, Icos, Ox40, and cRel. Regnase-1-deficient T cells show marked conversion into effector cells with elevated production of IFN-γ and IL-17A. Heterozygous Regnase-1 mice also show increased susceptibility to EAE, presumably because of hyperactivation of pathogenic T cells. Intriguingly, after the engagement of the T cell receptor (TCR), Regnase-1 is cleaved by Malt1, which is a protease component in the Card11-Bcl10-Malt1 (CBM) complex [148] (Figure 4d). Malt1 recognizes the serine–arginine sequence present in the N-terminus of Regnase-1 and results in cleavage after the P1 arginine residue. Interestingly, Malt1 protease dead mice displayed a mixed inflammatory phenotype involving multiple cell types [149,150,151]. In these mice, the generation of regulatory T cells was severely impaired, which led to autoimmune gastritis. Because Malt1 is also expressed in innate immune cells, Malt1 protease activity is required for the activation of DCs and NK cells via immunoreceptors such as Dectin-1/Dectin-2 and NKG2D/Ly49D, respectively [152].

Although evidence is very limited, Regnase-1 family proteins are also involved in inflammatory responses. Regnase-3 is highly expressed in myeloid lineages, such as macrophages and DCs, and localizes to endosome/phagosome [153]. At a younger age, major abnormalities in mice deficient in Regnase-3 are absent in the immune system. However, Regnase-3^–/–^ aged mice show mild lymphadenopathy, which is presumably due to sustained activation of IFNγ signaling in macrophages [153]. Nevertheless, endogenous target mRNAs for Regnase-3 are largely unknown. Similar to Regnase-1, the Regnase-3 protein undergoes proteasomal degradation in response to LPS [153], which suggests an unknown mechanism mediated through Regnase-1 and Regnase-3 regarding inflammation. Regnase-4 (also known as Zc3h12d or TFL) is also important in the suppression of T cell activation [154]. This gene shows lower expression in macrophages compared to lymphocytes, in particular, T cells. Mice with Regnase-4 deficiency showed severe EAE development, which is presumably due to sustained activation of pathogenic T cells [154]. Regnase-4 localizes to P bodies where Regnase-4 degrades mRNAs independent of Regnase-1 [154,155]. Based on their different cellular localizations, there are distinct mRNA decay mechanisms by Regnase proteins, although the precise mechanisms have not been elucidated.

Roquin also significantly contributes to the immune system. This molecule’s role was initially verified in mice that harbor a single point mutation present in the putative E3 ligase of Roquin, and these are called sanroque mice [156]. These sanroque mice displayed a lupus-like autoimmune disease, which was characterized by marked accumulation of follicular helper T (Tfh) cells and spontaneous germinal center formation. Icos mRNA was first documented as a Roquin target and explained Tfh cell expansion in sanroque mice [157,158]. Subsequently, Ifng mRNA and Ox40 mRNA were also shown to contribute to increased Tfh differentiation [159,160]. Similar to sanroque mice, T cell-specific deletion of both Rc3h1 and Rc3h2 causes spontaneous activation of T cells [160], which highlights the redundant roles of Roquin-1 and Roquin-2. Roquin proteins also regulate Th17 differentiation by controlling mRNA stability for Nfkbiz and Nfkbid [161]. It is also reported that Roquin is a substrate for Malt1 paracaspase [161], indicating that Malt1 acts as an ON/OFF switch for posttranscriptional mechanisms by controlling the expression levels of a set of mRNA decay regulators (Figure 4d). However, in macrophages, Roquin diminishes the expression of inflammatory cytokine genes such as Tnf and Il6. The role of Roquin in myeloid lineage cells was determined by analyzing sanroque mice under a RAG1 null background [162]. In an arthritis mouse model elicited by arthritogenic K/BxN serum, these mice developed severe inflammation in joints compared to control RAG1 knockout mice, which is mainly due to a major increase in TNF production.

## 8. Conclusions

RNA is involved in diverse biological processes and has a great impact on immune function, which enables innate-acquired interplay. Recently, there has been a significant advance in utilizing PRR-mediated RNA sensing. Intensive work on viral RNA recognition has provided new information on the molecular and structural basis of how TLRs and RLRs induce antiviral responses. Furthermore, nucleic acids are now considered potential adjuvants for therapeutic approaches. In parallel, greater attention is focused on the detrimental effects of PRR activation triggered by host RNAs, which is implicated in the development of autoimmunity. Recent work suggests that an altered threshold of activation of these RNA sensors or failure to maintain host RNAs, mostly non-coding RNAs, under control are keys to understanding how RNA sensors cause uncontrollable inflammation. Nevertheless, knowledge regarding tolerant mechanisms mediated by host RNA is very limited. Future investigations will be necessary to provide novel insights into non-coding RNAs as potential ligands of RNA sensors.

Another aspect of RNA regulation is the posttranscriptional control of gene expression that determines the protein output of inflammatory responses. Because there are as many as 1500 RBPs encoded in the genome [163], a complex network between RBPs and mRNAs exists to coordinate posttranscriptional processes. During the last decade, the CLIP technique has been improved and provides a significant insight into the molecular basis of RBP-RNA interaction [164]. RNA interactome studies show a proteome-wide picture of this interaction that is formed in vivo [165,166,167,168,169]. Diverse modifications on mRNAs also contribute to posttranscriptional regulation of immune responses [170]. These events seem to be changed by inflammatory signals quantitatively and qualitatively. Furthermore, there are many unconventional RBPs that do not harbor canonical RNA-binding domains. Therefore, it is still unclear how the RBP-RNA network globally modulates RNA metabolism. Future work will extend this field of research and provide a novel insight into new therapeutics for autoimmunity and other inflammatory diseases.

## Figures and Tables

**Figure 1 cells-09-01701-f001:**
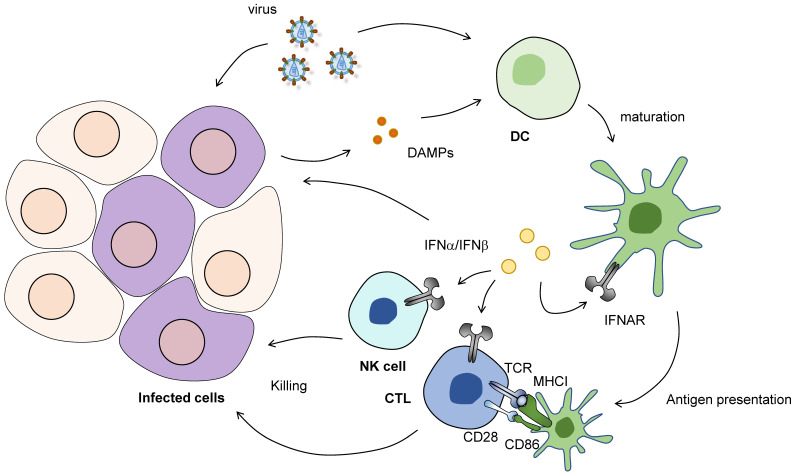
The role of type I IFNs between innate immunity and T cell-mediated antiviral immunity. Upon viral infection, direct recognition of viral nucleic acids by dendritic cells (DCs) and danger signals from infected cells can induce antiviral responses. When activated, DCs undergo maturation to express costimulatory molecules, such as CD86. In draining lymph nodes, these DCs act as antigen-presenting cells presenting viral antigens to CD8 T cells through dependence on MHC-I. Simultaneously, type I IFNs (e.g., IFNα and IFNβ) produced by innate immune and infected cells further activate CTLs and NK cells to enhance cytotoxicity. Type I IFNs also act on infected cells and neighboring cells, in an autocrine and paracrine fashion, which promotes apoptosis of infected cells and renders uninfected cells resistant to viral infection. CTL; cytotoxic T lymphocyte.

**Figure 2 cells-09-01701-f002:**
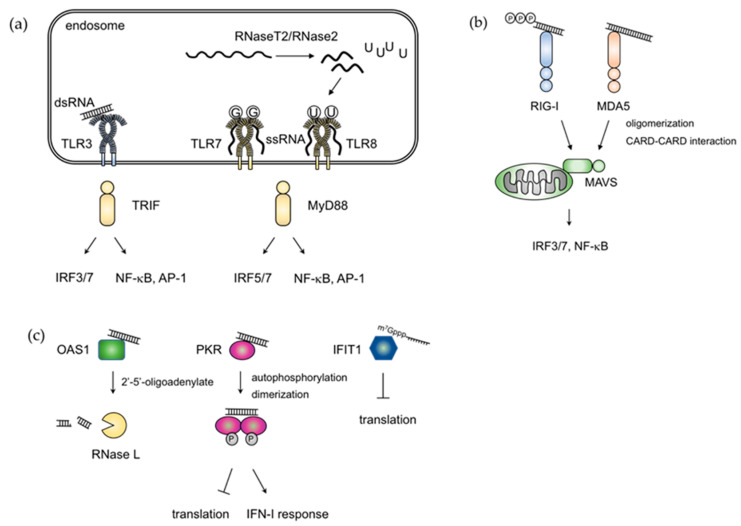
RNA sensing mechanisms. RNA sensing mechanisms are composed of endosomal and cytoplasmic sensors. (**a**) Endosomal RNA sensors, TLR3 and TLR7/8, recognize dsRNA and ssRNA, respectively. TLR3 activates the TRIF-dependent pathway to activate IRF3/7, NF-κB, and AP-1. In contrast, TLR7/8 triggers the MyD88-dependent pathway and leads to the activation of NF-κB and AP-1, and simultaneously activates IRF5/7 to induce type I IFN genes. In addition to recognizing ssRNA, TLR7 and TLR8 can be activated by guanosine and uridine residues, respectively. Uridine molecules for TLR8 are degraded products generated by RNaseT2 and RNase2, although it remains unknown how guanosine residues are generated. The engagement of both ligands achieves maximum activation of TLR7/8. (**b**) RIG-I and MDA5 are central RNA sensors that localize to the cytosol, and both harbor the N-terminal tandem caspase-recruitment domains (CARDs) and helicase domains. After binding to dsRNA, these RNA sensors undergo oligomerization along dsRNA structures, which then activates MAVS through CARD–CARD interaction. These sequential events then activate IRF3/7 along with NF-κB to induce IFN-I responses. (**c**) Alternative RNA sensing is mediated by RNA receptors whose expressions are induced by RLR activation. OAS1 generates 2′-5′-oligoadenylate that activates RNase L, which promotes the degradation of dsRNAs. The resulting fragments of dsRNAs can also activate the RIG-I-like receptor (RLR)-sensing pathway. PKR is a Ser/Thr protein kinase that is present in the unphosphorylated form. After binding to dsRNA, PKR undergoes autophosphorylation and dimerization, which inhibits viral protein synthesis and induces the IFN-I response. IFIT1 recognizes capped RNA with 5’-triphosphate ends, which can block the translation of viral RNA. U, uridine; G, guanosine; IFN-I, type I interferon; CARD, caspase activation recruitment domain.

**Figure 3 cells-09-01701-f003:**
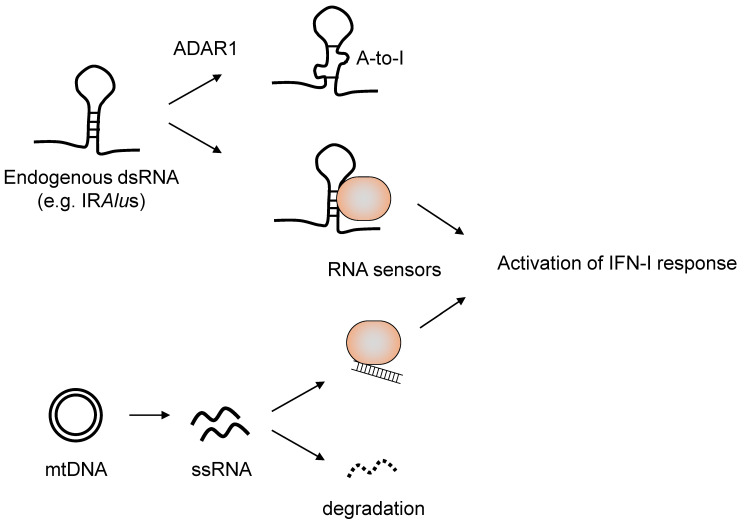
Mechanisms where endogenous RNAs can be recognized by RNA sensors. IRAlu elements can form dsRNA as self. Adenosine deaminase (ADAR) 1 destabilizes dsRNA structures via its A-to-I editing, which results in an escape from the recognition of host RNA sensors. mtDNA also potentially generates dsRNAs. The degradosome localized to the mitochondria can prevent dsRNA formation by limiting the accumulation of transcripts derived from mtDNA. In both cases, the abnormal formation of dsRNA can lead to aberrant type I IFN (IFN-I) responses. IRAlu; inverted repeat Alu, mtDNA; mitochondrial DNA.

**Figure 4 cells-09-01701-f004:**
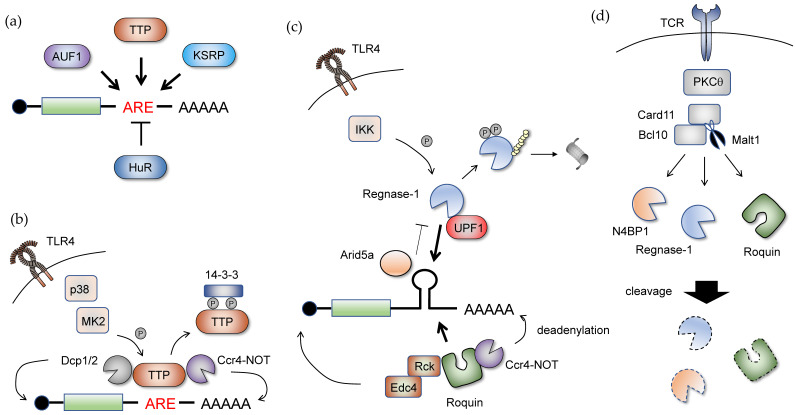
Posttranscriptional mechanisms by RNA-binding proteins (RBPs) regulating mRNA stability. (**a**) TTP, AUF1, and KSRP are representative RBPs that promote mRNA decay by binding to an AU-rich element (ARE) present in 3′UTR. HuR is an RBP that inhibits mRNA decay by competing for ARE sites. (**b**) TTP promotes mRNA decay through the recruitment of the deadenylation complex, CCR4-NOT, and Dcp1 and Dcp2 decapping enzymes. This can be suppressed by MK2-mediated phosphorylation of TTP, which is induced downstream of p38 mitogen-activated protein kinase (MAPK) in response to TLR4 activation. The phosphorylated form of TTP is sequestered by the 14-3-3 protein, which results in enhanced stability of target mRNAs. (**c**) Regnase-1 and Roquin are representative RBPs that recognize stem-loop structures present in 3′UTR. There are two distinct mRNA decay mechanisms: (1) Endonucleolytic cleavage by endonucleases and (2) mRNA decay induced by recruitment of the deadenylation complex and decapping enzymes. (1) Regnase-1 is an endonuclease that recognizes the stem-loop structure, and cooperatively promotes cleavage of target mRNA with UPF1. However, this process can be blocked by Arid5a. In addition, Regnase-1-mediated mRNA decay is also regulated when this protein undergoes IKK-mediated phosphorylation and subsequent proteasomal degradation. (**d**) Malt1 is a key component of the T cell receptor (TCR)-signaling pathway that contributes to posttranscriptional mechanisms. After the engagement of TCR, Malt1 is activated as a component of the Card11-Bcl10-Malt1 (CBM) complex downstream of PKCθ. Malt1 also acts on a set of substrates for its paracaspase, including Regnase-1, N4BP1, and Roquin. Malt1-mediated cleavage deactivates these proteins.

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
