# Peer review of "RNA Recognition and Immunity—Innate Immune Sensing and Its Posttranscriptional Regulation Mechanisms"

_cells, 2020, doi:10.3390/cells9071701_

Round 1
Reviewer 1 Report
Authors elegantly reviewed the host cell machinery to sense and distinguish self/non-self RNAs and elicite an adequate induction of IFN I-response. Also, authors brings up the role of RBPs that can guide the degradation of non-self RNA or control the mRNA stability and regulates inflammatory response. I recommend the publication of article after minor revision.
Minor points:
In section 3, authors can revisite the role of NLRP3 in recognition of viral RNA and inflammasome activation, as well as the role of RLRs in these signaling.
In the figure 2c, the representation of proteins (OAS1, PKR and IFIT1) as the same form and color leads the reader to misinterpretation of the function of each protein and characteristic of RNA that implicate in RNA-recognition by host cell. The figure can be improved.
Author Response
- In section 3, authors can revisit the role of NLRP3 in recognition of viral RNA and inflammasome activation, as well as the role of RLRs in these signaling.
We thank the reviewer for the kind suggestion. We agree that the role of NLRP3 in RNA recognition is an interesting field. However, in this manuscript, we prefer to focus on transcriptomic changes following RNA recognition by RLRs, and we think that NLRP3 and inflammasome activation is a bit outside the scope of this manuscript. Therefore, we would like to leave this aspect to other reviews.
- In the figure 2c, the representation of proteins (OAS1, PKR and IFIT1) as the same form and color leads the reader to misinterpretation of the function of each protein and characteristic of RNA that implicate in RNA-recognition by host cell. The figure can be improved.
We thank the reviewer for pointing out that Figure 2c is difficult to interpret in the original manuscript. According to the reviewer’s comment, we have modified Figure 2c to highlight each molecule represented in different colors and shapes (page 4).
Reviewer 2 Report
In this Review, Uehata T and Takeuchi O introduce the meaning of RNA-sensing and RNA-metabolism in the innate immune system. The authors well summarized information about known molecules involved in those phenomena. In general, this is a nice review.Considering that the authors showed various examples to show the role of RNA-metabolism in immune homeostasis in section 7, the authors are encouraged to introduce concrete examples suggesting the importance of PRRs in immune homeostasis, such as SLE induction by TLR7. Additionally, the following points listed below may help make this review refine.
Comments
- Line 58: Did the authors explain Human TLRs? Here, they should refer to species. In mice, TLR13 is also an RNA sensor.
- Lane 69-71, Figure2.(a): Under TLR7/9 activation, MyD88 signaling induce not only Inflammatory cytokines but also type-I IFN production via IRF5/7.
- Figure2.(a): TLR7 is a Guanosine sensor, but not Uridine sensor.
- Although, in the second part, the authors showed various examples suggesting the importance of RNA-metabolism in immune homeostasis, there are no concrete example suggesting the involvement of RNA-sensing by Pattern Recognition Receptors (PRRs) in immune homeostasis. Considering there are many examples indicating the involvement of PRRs in autoimmune or autoinflammation diseases, the authors are strongly encouraged to add new section introducing this point.
Author Response
- Line 58: Did the authors explain Human TLRs? Here, they should refer to species. In mice, TLR13 is also an RNA sensor.
We thank the reviewer for kind suggestions and pointing out that the species of the molecules described here in this manuscript was not clarified. As the reviewer pointed out, we focused on mammalian RNA sensing mechanisms, in particular in human. According to the reviewer’s comment, we have clarified the species of the molecules discussed in this manuscript. Also, we added the description of the function of mouse TLR13 (Abstract, page 1, line 19; Section-2, page 2, line 62-65).
- Lane 69-71, Figure2.(a): Under TLR7/9 activation, MyD88 signaling induce not only Inflammatory cytokines but also type-I IFN production via IRF5/7.
We agree that Figure 2a is potentially misleading. We have revised the Figure 2a (page 4).
Figure2.(a): TLR7 is a Guanosine sensor, but not Uridine sensor.
We agree with the reviewer to this point as well. Although we mentioned that TLR7 ligand is a guanosine molecule in the main text (page 5, line 128-129), the figure 2a was mistakenly depicted. We have revised the figure 2a and its figure legend to reflect this point.
Although, in the second part, the authors showed various examples suggesting the importance of RNA-metabolism in immune homeostasis, there are no concrete example suggesting the involvement of RNA-sensing by Pattern Recognition Receptors (PRRs) in immune homeostasis. Considering there are many examples indicating the involvement of PRRs in autoimmune or autoinflammation diseases, the authors are strongly encouraged to add new section introducing this point.
According to the reviewer’s suggestion, added a new paragraph in the section 4, following the statement on the diseases caused by abnormal cytosolic RNA sensors, in which statements of the roles of TLR 7 and TLR9 in autoimmunity has been added.
Reviewer 3 Report
The manuscript by Uehata and Takeuchi provides a review on the role of RNA-sensing mechanisms in the context of innate immunity. This is a well-written, easy to follow review, that I really enjoyed reading. It provides a clear and concise overview on the state of the art of this subject, with informative figures. I just have a few minor comments that in my opinion could help to further improve this manuscript.
Although the manuscript never explicitly states this, the authors mainly refer to human and vertebrate model organism. I believe this should be clearly stated upfront, as just some (but not all) of the RNA-sensing molecules described are shared by invertebrate metazoans lacking an innate immune system. This is a relevant information that needs to be added, as this is a hot topic of research on several commercially important invertebrates whose production is currently hampered by emerging viral infections.
Obviously, the authors do not need to include additional information to cover this particular aspect, as I realize RNA-sensing in invertebrates goes well beyond the scope of this review, but it is important that the subject of this MS is clearly identified for the readers’ benefit since the very beginning.
Similarly, it needs to be made clear upfront that the nomenclature of the molecules involved is based on human, as other vertebrates have a very different number of TLRs.
Figure 1 and others: make sure to remove line numbers from the figures in the final version
Concerning the activity of ADAR, maybe the authors might want to update some information by taking into account the possible RNA-modifying activity of this enzyme towards viral RNAs, as recently evidenced for example in SARS-CoV-2 by metatranscriptomics approach (see Di Giorgio et al. 2020, Science Advances).
Line 491: “Immune cells have evolved nucleic acid-sensing systems to combat pathogen invasion.”. I feel like this review lacks a brief statement somewhere concerning the evolutionary origins of such nucleic-acid sensing systems. Some of them are actually very ancient, and orthologous genes can be found throughout the entire metazoan tree of life. Others are also quite ancient, but orthology relationships are not equally easy to follow (e.g. TLRs, that have probably followed highly different evolutionary routes in verterbrates and invertebrates). Others are strictly confined to vertebrates and emerged along with the acquisition of an adaptive immune system.
Author Response
- Although the manuscript never explicitly states this, the authors mainly refer to human and vertebrate model organism. I believe this should be clearly stated upfront, as just some (but not all) of the RNA-sensing molecules described are shared by invertebrate metazoans lacking an innate immune system. This is a relevant information that needs to be added, as this is a hot topic of research on several commercially important invertebrates whose production is currently hampered by emerging viral infections.
Obviously, the authors do not need to include additional information to cover this particular aspect, as I realize RNA-sensing in invertebrates goes well beyond the scope of this review, but it is important that the subject of this MS is clearly identified for the readers’ benefit since the very beginning.
We thank the reviewer for the constructive suggestion. As the reviewer mentioned, this manuscript is mainly focused on mammalian species. According to this suggestion, we have clarified the species discussed in this manuscript (Abstract, page 1, line 19). Also, we added a new statement in Introduction (page 1, line 33-34), to explain that invertebrate RNA sensing system will be referred to other in-depth reviews.
- Similarly, it needs to be made clear upfront that the nomenclature of the molecules involved is based on human, as other vertebrates have a very different number of TLRs.
We agree with your suggestion. Because some of the molecules are overlapped between human and mouse, we have clarified that this manuscript is based on mammalian RNA sensing mechanisms (Abstract, page 1, line 19, Section-2, page 2, line 61-65).
- Figure 1 and others: make sure to remove line numbers from the figures in the final version
We have revised this point according to your comment.
- Concerning the activity of ADAR, maybe the authors might want to update some information by taking into account the possible RNA-modifying activity of this enzyme towards viral RNAs, as recently evidenced for example in SARS-CoV-2 by metatranscriptomics approach (see Di Giorgio et al. 2020, Science Advances).
We thank the reviewer for the suggestion. While ADAR1 may suppress viral propagation as one of the IFN-inducible genes, we think that the role of ADAR1 in viral infection has not been clarified. For instance, several works showed that ADAR1 prevents host antiviral responses. Therefore, we would like to leave this research area to other in-depth reviews. This explanation has been added to page 7, line 202-207 (Note that here we decided to compromise by adding a brief explanation to satisfy the reviewer, while at the same time avoiding these contradictory aspects of the function of ADAR1.)
- Line 491: “Immune cells have evolved nucleic acid-sensing systems to combat pathogen invasion.”. I feel like this review lacks a brief statement somewhere concerning the evolutionary origins of such nucleic-acid sensing systems. Some of them are actually very ancient, and orthologous genes can be found throughout the entire metazoan tree of life. Others are also quite ancient, but orthology relationships are not equally easy to follow (e.g. TLRs, that have probably followed highly different evolutionary routes in verterbrates and invertebrates). Others are strictly confined to vertebrates and emerged along with the acquisition of an adaptive immune system.
We agree with your point. As the review pointed out, this manuscript does not focus on the evolutionary aspect of RNA sensing system between species. Related to comment-#1, as to this point, we would like to refer to other in-depth reviews. Besides, to prevent confusion, we have removed the sentence, “Immune cells have evolved nucleic acid-sensing systems to combat pathogen invasion.”